# Dysregulation of the Enteric Nervous System in the Mid Colon of Complement Component 3 Knockout Mice with Constipation Phenotypes

**DOI:** 10.3390/ijms23126862

**Published:** 2022-06-20

**Authors:** Yun Ju Choi, Hee Jin Song, Ji Eun Kim, Su Jin Lee, You Jeong Jin, Yu Jeong Roh, Ayun Seol, Hye Sung Kim, Dae Youn Hwang

**Affiliations:** 1Department of Biomaterials Science (BK21 FOUR Program) and Life and Industry Convergence Research Institute, College of Natural Resources & LifeScience, Pusan National University, Miryang 50463, Korea; poiu335@naver.com (Y.J.C.); hejin1544@naver.com (H.J.S.); prettyjiunx@naver.com (J.E.K.); nuit4510@naver.com (S.J.L.); hjinyuu1@naver.com (Y.J.J.); buzyu99@naver.com (Y.J.R.); a990609@naver.com (A.S.); 2Department of Nanomechatronics Engineering, College of Nanoscience & Nanotechnology, Pusan National University, Miryang 50463, Korea; hsk@pusan.ac.kr; 3Laboratory Animals Resources Center, Pusan National University, Miryang 50463, Korea

**Keywords:** C3, constipation, enteric nervous system, myenteric neuron, ICC, 5-HT, Ach, NO

## Abstract

Complement component 3 (C3) contributes to neurogenesis, neural migration, and synaptic elimination under normal and disease conditions of the brain, even though it has not been studied in the enteric nervous system (ENS). To determine the role of C3 in the regulatory mechanism of ENS during C3 deficiency-induced constipation, the changes in the markers of neuronal and interstitial cells of Cajal (ICCs), the markers for excitatory and inhibitory transmission of ENS, and expression of C3 receptors were analyzed in the mid colon of C3 knockout (KO) mice at 16 weeks of age. Prominent constipation phenotypes, including the decrease in stool parameters, changes in the histological structure, and suppression of mucin secretion, were detected in C3 KO mice compared to wildtype (WT) mice. The expression levels of the neuron specific enolase (NSE), protein gene product 9.5 (PGP9.5), and C-kit markers for myenteric neurons and ICCs were lower in the mid colon of C3 KO mice than WT mice. Excitatory transmission analysis revealed similar suppression of the 5-hydroxytryptamine (5-HT) concentration, expression of 5-HT receptors, acetylcholine (ACh) concentration, ACh esterase (AChE) activity, and expression of muscarinic ACh receptors (mAChRs), despite the mAChRs downstream signaling pathway being activated in the mid colon of C3 KO mice. In inhibitory transmission analysis, C3 KO mice showed an increase in the nitric oxide (NO) concentration and inducible nitric oxide synthase (iNOS) expression, while neuronal NOS (nNOS) expression, cholecystokinin (CCK), and gastrin concentration were decreased in the same mice. Furthermore, the levels of C3a receptor (C3aR) and C3bR expression were enhanced in the mid colon of C3 KO mice compared to the WT mice during C3 deficiency-induced constipation. Overall, these results indicate that a dysregulation of the ENS may play an important role in C3 deficiency-induced constipation in the mid colon of C3 KO mice.

## 1. Introduction

C3, the most abundant complement protein, plays an important role in activating the complement system during the formation of membrane attack complex (MAC) to lyse the target pathogens [1,2]. In the complement activation process, this protein is cleaved into C3a and iC3b and C3f by C3 convertase via the classical, alternative, and lectin pathways [3]. The resulting fragments derived from C3 proteins bind to several receptors, such as CR1, CR2 (CD21), CR3 (CD11b/CD18), and CR4 (CD11c/CD18), and stimulate various immune responses, including phagocytosis, solubilization of the immune complex, and neutralization of viruses via direct recognition and interaction [4]. In addition, C3 proteins help regulate development in the normal and pathological brain. This role is associated with protein expression in most brain cells, including astrocytes, microglia, oligodendrocytes, and neurons, during the classical and alternative pathway, even though the primary source of C3 is astrocytes [5,6]. In these cells, C3 fragments and their receptors mediate neurogenesis, neuronal migration, and synaptic pruning [7,8,9]. Furthermore, C3 proteins have been implicated in a few brain disorders. Synaptic terminal elimination induced by a neuroinvasive infection with West Nile Virus was decreased significantly in C3- and C3R-deficient mice [10]. The synaptic loss and neuronal loss, as well as microglial activation, are positively correlated with increased levels of C3 and C3aR expression [11,12]. On the other hand, the above functions of C3 and their receptors have not been fully studied in the nervous system of other organs except the brain.

Indirect evidence first suggested the role of C3 in constipation. The level of C3 mRNA was upregulated in inflammatory bowel disease (IBD) and Crohn’s disease (CD) patients [13,14]. Similar increases in the concentrations of C3 and C3b fragments were observed. An intensive level of C3 proteins was detected in the small and large intestine in CD patients, ulcerative colitis (UC) patients, and dextran sodium sulfate-treated mice [15,16,17,18,19]. Recently, the potential of C3 as a cause of constipation was proven using C3 KO mice. Sixteen week old C3 KO mice exhibited significant changes in the constipation phenotypes, including the stool excretion parameters, gastrointestinal transit, intestine length, histological structure of the colon, and mucin secretion during C3 deficiency [20]. In addition, C3 deficiency-induced constipation of these mice was associated with dysbiosis of 12 genera of bacteria among the total fecal microbiota [21]. Furthermore, significant inflammatory responses were induced during C3 deficiency in the mid colon of C3 KO mice. This response was mediated by activating the apoptosis-associated speck-like protein containing a caspase recruitment domain (ASC) inflammasome pathway, NF-κB signaling pathway, and iNOS-mediated COX-2 induction pathway [22]. On the other hand, whether C3 deficiency-induced constipation can be accompanied by a dysfunction of the ENS in the mid colon of C3 KO mice remains to be determined.

This study examined alterations on markers associated with excitatory and inhibitory transmission of the ENS in the mid colon of C3 KO mice with the constipation phenotypes. In particular, this study focused on the markers for ICC and neuron cells, markers for the excitatory and inhibitory transmission in ENS, and the mAChR downstream signal pathway.

## 2. Results

### 2.1. Confirmation of the Constipation Phenotypes in C3 KO Mice

In a previous study, 16 week old C3 KO mice exhibited constipation phenotypes, including a decrease in excretion parameters, disruption of the colon histological structure, and decline in mucin secretion ability [20]. Therefore, these phenotypes were first confirmed in 16 week old C3 KO mice before analyzing the dysregulation of ENS in the mid colon. Three stool parameters, including the number, weight, and water content, were decreased remarkably in the C3 KO mice compared to those of WT mice. The water volume lost per hour in stools from C3 KO mice (0.225 mL/h) was less than 15% of that lost from WT mice (1.52 mL/h) (Figure 1a). The cause of this difference is considered to be related to the decrease in water intake at 14–16 weeks of age because the urine volume was constantly maintained (Appendix A). Similar alterations were observed on the histological structure, including the thickness of the mucosal layer and muscle layer and mucin secretion of the mid colon (Figure 1b,c). These results suggest that 16 week old C3 KO mice show constant phenotypes for chronic constipation, allowing an investigation of the ENS dysfunction.

### 2.2. Alteration of the Makers for Neuronal Cells and ICC in the Mid Colon of C3 KO Mice

The expression levels of the markers, including C-kit, NSE, and PGP9.5 proteins, were measured in the mid colon of C3 KO mice to indirectly determine if C3 deficiency-induced constipation was accompanied by the decreased density of neuronal cells and ICC in the mid colon of C3 KO mice. The expression levels of three proteins were significantly lower in the mid colon of C3 KO mice than WT mice. The largest decrease was detected in the expression level of C-kit, followed by PGP9.5 and NSE (Figure 2a). Furthermore, the distribution of PGP9.5 proteins in the morphometric analysis of the neural population was observed to confirm whether the results of Western blot were well reflected. The PGP9.5-stained subpopulation was lower in the mid colon of C3 KO mice than in WT mice. Furthermore, they were majorly packed between muscle layers, while weak intensity of fluorescence was observed between submucosal layer and muscle layer. Therefore, fewer PGP9.5 proteins in Western blot were considered to be associated with a decrease in muscle layer (Figure 2b). Overall, the above results show that C3 deficiency-induced constipation may be tightly linked to the decrease in the markers for neuronal cells and ICC in the mid colon of C3 KO mice.

### 2.3. Changes in the Excitatory Function of ENS in the Mid Colon of C3 KO Mice

The regulatory factors for 5-HT and ACh function were analyzed in the mid colon of C3 KO mice with constipation to determine if C3 deficiency-induced constipation was accompanied by an alteration on the excitatory function of ENS in the mid colon. The concentration of 5-HT significantly decreased by 18% in C3 KO mice compared to WT mice (Figure 3a). In addition, a similar decrease was detected in four types of 5-HT receptors, namely, 2AR, 2BR, 3AR, and 3BR, even though their rate of decrease was varied. These rates were greater in the levels of 2BR and 3BR expression than those of 2AR and 3AR (Figure 3b,c). The concentration of ACh was lower in the mid colon of C3 KO mice than the wildtype mice, while the activity of AChE was remarkably higher in the same sample (Figure 4a,b). A similar decreasing pattern was observed in the expression of mAChRs. These mAChR M2 and M3 levels were remarkably lower in the mid colon of the C3 KO mice than the WT mice (Figure 5a). On the other hand, a reverse pattern was detected for the expression of a key member in the signaling pathway of mAChR downstream. The levels of Gα expression and PKC and PI3K phosphorylation were significantly higher in the mid colon of C3 KO mice than the WT mice (Figure 5b,c). These signals were ultimately reflected in the level of MLC phosphorylation, but the rate of the increase was different (Figure 5c). These results suggest that C3 deficiency-induced constipation may be tightly linked to a dysregulation of the excitatory function of ENS in the mid colon of C3 KO mice.

### 2.4. Change in the Inhibitory Function of ENS in the Mid Colon of C3 KO Mice

This section investigates whether C3 deficiency-induced constipation was accompanied by an alteration in the inhibitory function of ENS in the mid colon. The regulatory factors for NO functions were analyzed in the mid colon of C3 KO mice. The NO concentration was remarkably higher at 41% in C3 KO mice than WT mice (Figure 6a). A similar increase pattern was detected in the expression of iNOS protein, but the magnitude of the increase was different (Figure 6b). By contrast, nNOS expression showed a reverse pattern to iNOS expression (Figure 6b). Therefore, C3 deficiency-induced constipation may be associated with a dysregulation of the excitatory function of ENS through upregulation of the NO concentration and iNOS expression in the mid colon of C3 KO mice.

### 2.5. Alterations on the Concentration of GI Hormones in the Mid Colon of C3 KO Mice

ENS plays a key role in the secretion of gastrointestinal hormones, including CCK and gastrin, to control the digestive process, such as gut motility, enzyme secretion, and energy homeostasis [23]. The concentrations of CCK and gastrin hormone were measured in the mid colon of C3 KO mice to determine if the dysregulation of ENS function was accompanied by a change in the regulation of GI hormones in the mid colon of C3 KO mice. These levels were remarkably lower in the mid colon of C3 KO mice than in WT mice (Figure 7). Hence, dysregulation of the ENS function is associated with the downregulation of GI hormones in the mid colon of C3 KO mice.

### 2.6. Verification of the Correlation between ENS Function and C3R

Alteration of C3R expression is associated with synaptic loss and microglial activation in the brain of immune disorders with West Nile virus infection and neurodegenerative disorder model [9,12]. Hence, we examined the expression of C3Rs in the colon tissue of C3 KO mice to verify the involvement of C3R during dysregulation of ENS. The levels of C3aR and C3bR expression were remarkably higher in the mid colon of C3 KO mice (47% and 82%, respectively) than WT during complete downregulation of the C3 proteins (Figure 8). Therefore, these results suggest that C3aR and C3bR may be associated with dysregulation of the ENS function in the mid colon of C3 KO mice.

## 3. Discussion

A dysfunction of ENS can be considered one of the causes of constipation, reflux, and delayed gastric emptying because it plays a key role in regulating the GI tract motility, movement of water and electrolyte, gastric and pancreatic secretion, and defense reactions [24,25,26]. Therefore, it is very important to verify the dysfunction of ENS in the C3 KO mice as a new constipation model. This study examined whether C3 deficiency-induced constipation can be accompanied by a dysfunction of the ENS in the mid colon of C3 KO mice. These results provide the first evidence that C3 deficiency-induced constipation may be associated with an abnormality in the density of neurons and ICC in ENS, excitatory and inhibitor function of the ENS, and C3R expression in the mid colon of C3 KO mice.

The gut motility is regulated by a complex collaboration and communication of various ENS-related cells, including enteric neurons, ICC, and smooth muscle [27]. During the pathogenesis of constipation, significant alterations were detected in the number of enteric neurons and ICC in animal models and human patients. Slow-transit constipation (STC) patients showed a significant decrease in the number of ICC and PGP9.5 reactive cells of the sigmoid colon or across the whole colon [28,29,30]. In addition, the number of enteric neurons was lower in the submucosal plexus of patients with intractable constipation [31]. Furthermore, a similar decrease pattern was detected in constipated animal models. The levels of NSE, PGP9.5, and C-kit expression were statistically significantly lower in the mid colon of 16 week old Lep KO mice with the constipation phenotypes [32]. ICR mice and SR rats with loperamide (Lop)-induced constipation exhibited a decrease in the level of two ICC markers (C-kit and SCF) and the other enteric nerve-related factors (PGP9.5, brain-derived neurotrophic factor (BDNF), and GNDT) [33,34]. This study measured the expression levels of NSE, PGP9.5, and C-kit markers in the mid colon of C3 KO mice to indirectly determine if C3 deficiency-induced constipation is accompanied by changes in the density of enteric neurons and ICC. Most results from the current study for the specific markers of enteric neurons and ICC in C3 KO mice were similar to previous studies that directly examined the number of the enteric neuron and ICC in patients and animal models with the constipation phenotypes. However, the above comparisons have limitations because our study did not directly count the numbers of two cells and measured them indirectly using specific markers. Furthermore, it should be fully considered that the amount of C-kit protein does not absolutely reflect the amount of ICC because it is a marker of other than ICC. Therefore, further studies will be needed to directly count and identify the number of enteric neurons and ICC through full immunohistochemical characterization and quantification using Western blot.

The present study investigated alterations in the excitatory function of ENS in the mid colon of C3 KO mice during C3 deficiency-induced constipation. The concentration of two neurotransmitters (5-HT and ACh) and the expression level of their associate receptors were remarkably lower in the mid colon of C3 KO mice than the WT mice, as shown in Figure 3, Figure 4 and Figure 5. Generally, the motility and transit of the GI tract are regulated spatiotemporally by ENS activities using several neurotransmitters, such as 5-HT and ACh. During constipation, the concentrations of these neurotransmitters were changed statistically significantly in human patients and rodent models. The 5-HT concentration was reduced significantly in patients with functional constipation (FC), constipation-predominant irritable bowel syndrome during normal feeding, and alternating diarrhea and constipation symptoms [35,36,37]. A similar decrease pattern in the 5-HT concentration was observed in the serum of Lop-induced constipation SD rats and the constipation population fecal fluid gavage group, while the expression level of the 5-HT receptor was lower in the distal colon of Lop-induced STC mice than WT mice [38,39,40,41]. On the other hand, only a few opposite results were reported in patients. The overall mean 5-HT concentrations in the colonic mucosal specimens were higher in patients with constipation-predominant irritable bowel syndrome (IBS) than the control subjects and patients with diarrhea-predominant IBS [42]. Overall, these results are consistent with previous results that detected 5-HT concentrations in patients, rats, and mice, even though there are differences in the causes of constipation. Significant alterations in ACh concentration, AChE activity, and AChR downstream signaling were detected. Lop-induced constipation rat and CRISPR/Cas9-generated Lep KO mice with obesity-induced constipation exhibited a decrease in mAChRM2 and M3 expression level, as well as activation of their downstream signaling [32,43,44]. In addition, the AChE activity was statistically significantly lower in the colon tissue of Lop-induced constipation rats and the serum of activated carbon-induced constipation mice than their untreated counterparts [44,45]. These findings are consistent with previous studies in mAChR expression and their downstream signaling pathway, but AChE activity tended to be opposite in previous studies. The difference between the results of the present study and previous studies was attributed to differences in the cause of constipation because there are many differences in the types of causative proteins or substances that induce this disease.

NO is one of the major non-andrenergic noncholinergic (NANC) inhibitory neurotransmitters that relax the GI smooth muscle to regulate physiologic peristalsis [46,47,48]. A disruption of the NO concentration can alter normal GI motility, but this molecule is produced by nNOS in the gut [49,50]. In particular, the pathological phenotypes of constipation were tightly linked to the changes in the NO contents and NOS expression. NO nerves strongly innervated in the colons of patients with STC compared to those of normal individuals [51]. The level of NOS mRNA was significantly higher in the colon of STC rats established by gastric irrigation of rhubarb and the small intestine of activated carbon-induced constipated mice [52,53]. In addition, the NO and NOS concentrations in plasma were increased in the Lop-induced functional constipation rats [54]. Atropine–diphenoxylate-induced STC rats showed higher NO and NOS values in the serum than normal rats [53]. This study examined whether C3 deficiency-induced constipation may be accompanied by NO-mediated inhibitory function of ENS in the mid colon. The NO concentration and nNOS expression were lower in the mid colon of C3 KO mice, while iNOS expression was higher in the same mice. The NO concentration increased similarly in this study using C3 KO mice and in previous studies using Lop-induced rats. On the other hand, there was a difference in the results of the NOS mRNA levels. This study revealed an increase in iNOS and a decrease in nNOS, where previous studies reported an increase in the NOS mRNA level regardless of their type. This disparity results from differences in the causes of constipation, tissue samples analyzed, and the subtypes of NOS detected.

Lastly, the C3 receptors, including CR1, CR2, CR3, and CR4, contribute to the migration and phagocytosis of cells and immune regulation through binding with several C3 fragments, such as C3a, iC3b, C3d, and C3f [54,55]. Significant alterations in the expression of these receptors have been detected in various diseases, such as inflammatory diseases [56,57], vascular disease [58], and microbial infection, even though C3bα and β-protein were maintained consistently in several strains of mice [59,60]. On the other hand, these changes in constipation conditions have been examined in limited studies. The levels of C3 expression were increased significantly, whereas expressions of C3aR and C3bR were decreased during Lop-induced constipation [61]. The present study investigated the levels of C3a and C3bR expression in the mid colon of C3 KO mice during C3 deficiency-induced constipation. These mice showed the upregulation of C3aR and C3bR expression during the complete downregulation of the C3 protein. The results of the present study in C3 KO mice tended to be opposite those of Lop-induced constipation rats. This is thought to be due to the differences in the action mechanism that induces the constipation phenotypes. Additional studies on the transcriptional regulation mechanisms of C3R during two constipation induction process will be helpful in the study to identify these differences. Therefore, these findings provide additional evidence for the close interactions between constipation and C3Rs expression, but more studies will be needed to explain the clear difference in both studies. Furthermore, our study had some limitations in that it did not examine clinical observations due to the fact that human disease etiopathogenesis is much more complex. Moreover, the lack of any comparison in relation to other animal models with similar phenotypes and sex specificity, as well as data normalization using only one housekeeping gene, should be considered as a drawback of our study.

## 4. Materials and Methods

### 4.1. Management and Breeding of C3 KO Mice

The protocol for the animal study was approved by the Pusan National University-Institutional Animal Care and Use Committee (PNU-IACUC; Approval Number PNU-2020-2657). All mice were handled at the Pusan National University Laboratory Animal Resources Center, which is accredited by the Korea Food and Drug Administration (FDA) (Accredited Unit Number: 000231) and AAALAC International (Accredited Unit Number: 001525). They were provided with a standard irradiated chow diet (Samtako BioKorea Co., Osan, Korea) ad libitum consisting of moisture (12.5%), crude protein (25.43%), crude fat (6.06%), crude fiber (3.9%), crude ash (5.31%), calcium (1.14%), and phosphorus (0.99%). The animals were maintained in a specific pathogen-free (SPF) state under a strict light cycle (lights on at 6:00 a.m. and off at 6:00 p.m.) at 22 ± 2 °C and relative humidity of 50% ± 10%.

Eight week old C3 KO mice with CRISPR/Cas9-mediated C3 mutant genes and WT FVB/N background strains were kindly provided by the Department of Laboratory Animal Resources (Laboratory Animals Resources Bank) at the National Institute of Food and Drug Safety Evaluation (NIFDS, Chungju, Korea). The KO mice were confirmed by DNA- Polymerase Chain Reaction (PCR) analysis of the genomic DNA using the specific primers described elsewhere [20]. Sixteen week old WT (*n* = 7) and KO (*n* = 7) mice were bred in a metabolic cage to analyze the excretion parameters. Moreover, the total sample size of 14 was calculated using G-POWER 3.1.9.7 software in order to determine the sufficient number of mice per group. Subsequently, they were euthanized in a chamber filled with CO_2_ gas. The mid colon tissue was selected on the basis of its role in the peristalsis, defecation, water absorption, and stool formation, and it was collected for further analysis, including histological structure, Western blot, Quantitative Real-Time PCR (qRT-PCR), and Enzyme-linked immunosorbent assay (ELISA).

### 4.2. Stool Parameter Analysis

Briefly, WT (*n* = 7) and C3 KO (*n* = 7) mice were bred individually in metabolic cages (Daejong Instrument Industry Co., LTD, Seoul, Korea) for 12 h to obtain uncontaminated stool samples. After collecting the total stools, their weights were measured three times using an electronic balance, while the number was counted twice. The moisture content of stool was calculated as follows:Stool moisture content = (A − B)/A × 100,(1)
where A is the weight of fresh stools immediately collected from mice, and B is the weight of stools after drying at 60 °C for 12 h.

### 4.3. Histopathological Analysis

The mid colon tissues collected from WT (*n* = 5) and C3 KO (*n* = 5) mice were fixed in 10% formalin for 48 h, embedded in paraffin wax, and sectioned into 4 μm thick slices that were stained with hematoxylin and eosin (H&E; Sigma-Aldrich Co., St. Louis, MO, USA). Finally, morphological structures of these sections were observed at 100× and 400× magnification under an optical microscope (Leica Microsystems, Herbrugg, Switzerland).

The level of PGP9.5 was detected by immunofluorescence staining analysis. The mid colon tissues of mice (*n* = 3) in subset group were fixed in 10% formalin for 48 h, embedded in paraffin blocks, and sliced into 4 μm thick sections. The sections (*n* = 5) were then deparaffinized with xylene, rehydrated with different EtOH concentrations, and pretreated with a blocking buffer containing 10% goat serum (Vector Laboratories, Inc., Burlingame, CA, USA) in a 1× PBS solution for 30 min at room temperature. The pretreated sections were then incubated with anti-PGP9.5 (Abcam Com., Cambridge, UK; ab108986) antibodies diluted 1:200 in a blocking buffer. After thorough washing in 1× PBS solution, the sections were incubated with goat fluorescein isothiocyanate (FITC)-labeled anti-rabbit IgG (1:200, Thermo Fisher Scientific Inc., Wilmington, MA, USA) for 45 min, washed three times in 1× PBS for 30 min each, and mounted with a vector shield mounting medium. Finally, the green fluorescence intensity on the tissue section of the mid colon was detected using a Motic AE31 Inverted Phase Contrast Fluorescence Microscope (Motic Incorporation Ltd., Causeway Bay, Hong Kong, China).

Mucin-staining analysis was achieved by fixing the mid colons collected from the mice (*n* = 3) of all subset groups in 10% formalin for 48 h, embedding the samples in paraffin wax, and sectioning them into 4 μm thick slices that were then deparaffinized with xylene and rehydrated. The mounted tissue sections were rinsed with distilled water and stained using an Alcian blue stain kit (IHC WORLD, Woodstock, MD, USA). The stained patterns in the mid colon sections were observed by optical microscopy (Leica Microsystems).

### 4.4. Western Blotting Analysis

The total tissue proteins were collected from the mid colons of WT (*n* = 4) and C3 KO (*n* = 4) mice using a Pro-Prep Protein Extraction Solution (Intron Biotechnology Inc., Seongnam, Korea), according to the manufacturer’s protocol. The acquired proteins were then centrifuged at 13,000 rpm at 4 °C for 5 min, after which the total protein concentrations were determined using a SMARTTM Bicinchoninic Acid Protein assay kit (Thermo Fisher Scientific Inc.). The proteins (30 μg) were subjected to 4–20% sodium dodecyl sulfate polyacrylamide gel electrophoresis (SDS-PAGE) for 3 h. The resolved proteins were transferred to nitrocellulose membranes for 2 h at 40 V. The membranes were then probed with the following primary antibodies overnight at 4 °C: anti-Gα (Abcam Com.; ab283266), anti-mAChR M2 (Alomone Labs, Jerusalem, Israel; AMR-002), anti-mAChR M3 (Alomone Labs; AMR-006), anti-PKC (Cell Signaling Technology Inc., Cambridge, MA, USA; #2056), anti-p-PKC (Cell Signaling Technology Inc.; #9375), anti-PI3K (Cell Signaling Technology Inc.; #4292), anti-p-PI3K (Cell Signaling Technology Inc.; #4228), anti-MLC (Abcam Com.; ab79935), anti-p-MLC (Abcam Com.; ab2480), anti-nNOS (Abcam Com.; ab3511), anti-NSE (Abcam Com.; ab79757), anti-C-kit (DAKO, Kyoto, Japan; A4502), anti-PGP9.5 (Abcam Com.; ab108986), or anti-actin (Cell Signaling Technology Inc.; #4967). The membranes were then washed with a washing buffer (137 mM NaCl, 2.7 mM KCl, 10 mM Na_2_HPO_4_, 2 mM KH_2_PO_4_, and 0.05% Tween-20), followed by incubation with 1:1,000 diluted horseradish peroxidase-conjugated goat anti-rabbit IgG (Zymed Laboratories, South San Francisco, CA, USA) for 2 h at room temperature, after which the blots were developed using a Chemiluminescence Reagent Plus kit (Pfizer Inc., Gladstone, NJ, USA). The signal images of each protein were then acquired using a digital camera (1.92 MP resolution) of the FluorChem^®^ FC2 Imaging system (Alpha Innotech Corporation, San Leandro, CA, USA). The protein densities were semi-quantified using the AlphaView Program, version 3.2.2 (Cell Biosciences Inc.).

### 4.5. qRT-PCR Analysis

The total RNA molecules were isolated from the mid colon of mice (*n* = 4) in subset group using an RNA Bee solution (Tet-Test, Friendswood, TX, USA). After homogenizing the frozen mid colon tissue, RNA molecules were isolated by centrifugation at 15,000 rpm for 15 min, and their concentration was measured using the Nano Drop Spectrophotometer (Allsheng, Hangzhou, China). The total RNA (5 µg) from the mid colon tissue was annealed with 500 ng of oligo-dT primer (Thermo Fisher Scientific Inc.) at 70 °C for 10 min to examine the transcription level of each gene. The complementary DNA (cDNA) was synthesized using the Invitrogen Superscript II reverse transcriptase (Thermo Fisher Scientific Inc.). qRT-PCR was performed using the cDNA template obtained (2 µL) and 2× Power SYBR Green (6 µL; Toyobo Life Science, Osaka, Japan) containing the following specific primers: 5-HT 2AR, sense primer 5′–CCGGG AGCCT CTTGA TACAG–3′ and antisense primer 5′–AGCCC CTCTC AAAGT CACAC A–3′; 5-HT 2BR, sense primer 5′–GCAGA TTTGC TGGTT GGATT G–3′ and antisense primer 5′–GGCCA TATAG CCTCA AACAT GAT–3′; 5-HT 3AR (5-hydroxytryptamine receptor 3A) sense primer 5′–CTGAG GCCCT CCCAC ATCT–3′ and antisense primer 5′–GGAAA GGAAC AAGGC CAACA–3′; 5-HT 3BR (5-hydroxytryptamine Receptor 3B) sense primer 5′–TGCCG AGGAG TCTAG ATTGT ACCT–3′ and antisense primer 5′–ACCCG ATGCT CCTGA TGGA–3′; β-actin sense primer 5′–ACGGC CAGGT CATCA CTATT G–3′ and antisense primer 5′–CAAGA AGGAA GGCTG GAAAA GA–3′. qRT-PCR was performed for 40 cycles using the following sequence: denaturation at 95 °C for 15 s, followed by annealing and extension at 70 °C for 60 s. The fluorescence intensity was measured at the end of the extension phase of each cycle. The threshold value for the fluorescence intensities of all samples was set manually. The reaction cycle at which the PCR products exceeded this fluorescence intensity threshold during the exponential phase of PCR amplification was considered the threshold cycle (Ct). The expression of the target gene was quantified relative to that of the housekeeping gene β-actin on the basis of a comparison of the Cts at constant fluorescence intensity, as per Livak and Schmittgen’s method [18]. The relative quantification formula was the 2^−ΔΔCt^ method proposed by Livak and Schmittgen. In C3 KO mice, the Ct value of target gene mRNA was normalized (Ct target gene − Ct β-actin) as the β-actin mRNA Ct value, which was called ΔCt. The Ct_calibrator_ (mean target gene − mean β-actin) was calculated by calculating the average of the target gene mRNA Ct value and the β-actin mRNA Ct value of WT groups (*n* = 4). With this ΔCt_calibrator_, the ΔCt of C3 KO mice was normalized again, which was called ΔΔCt (ΔCt − ΔCt_calibrator_). This ΔΔCt was expressed as a final relative quantitative value of 2^−ΔΔCt^.

### 4.6. Measurement of 5-HT Concentrations

The concentration of 5-HT was quantified in the mid colon mice (*n* = 5) in subset group using ELISA kits (ImmuSmol, Pessac, France) according to the manufacturer’s instructions. Briefly, the mid colon tissues (20 mg) were homogenized in ice-cold 1× PBS (pH 7.2–7.4) using a glass homogenizer (Sigma-Aldrich Co.). The resulting tissue lysates were then centrifuged at 10,000× *g* for 10 min at 4 °C, after which the supernatant was collected for analysis. First, diluted standards and samples were acylated for 30 min at room temperature (RT) on a shaker, after which serotonin antiserum was added to the acylated standards and samples for 15–20 h at 2–8 °C. After conjugating enzyme in each well for 30 min at RT, the substrate reagent was added and incubated further for 30 min at RT. Finally, the reaction was then quenched using the stop solution, and the absorbance of the reaction mixture was read at 450 nm using the VersaMax Plate Reader (Molecular Devices, Sunnyvale, CA, USA).

### 4.7. Determination of ACh Concentration

The concentration of ACh was quantified using an Acetylcholine Assay Kit (Cell Biolabs Inc., San Diego, CA, USA) in accordance with the manufacturer’s protocols. Briefly, the mid colon from mice (*n* = 5) in subset group was homogenized in ice-cold 1× PBS (pH 7.2–7.4) using a glass homogenizer (Sigma-Aldrich Co.); the homogenate was stored at −70 °C prior to ACh analysis. For analysis, the homogenate sample (or standards) and the ACh reaction mixture were incubated on an orbital rotator for 45 min at RT in a 96-well plate protected from light. Color alteration within the plate wells was determined using a Vmax plate reader (Molecular Devices) at 570 nm (excitation) and 600 nm (emission).

### 4.8. Determination of AChE Activity

The level of AChE activity was determined using an Acetylcholinesterase Assay Kit (Abcam Com.) in accordance with the manufacturer’s protocols. Briefly, the mid colon tissue from mice (*n* = 5) in subset group was homogenized in ice-cold 1× PBS (pH 7.4); the homogenate was stored at −70 °C prior to AChE analysis. For analysis, the homogenate sample (or standards) and the AChE reaction mixture were incubated for 20 min at RT in a 96-well plate protected from light. Color alteration within the plate wells was determined using a Vmax plate reader (Molecular Devices) at 405 nm.

### 4.9. Determination of the NO Concentration

The level of nitrite, which is the stable reaction product generated from NO with molecular oxygen, was used to indicate NO production. The mid colon tissue (20 mg) from mice (*n* = 5) in subset group was homogenized in ice-cold 1× PBS (pH 7.4) and centrifuged at 14,000 rpm for 5 min at 4 °C. The supernatants of the tissues were collected from mid colon and kept at −80 °C until use. Duplicates of 100 μL of the supernatant of homogenate were added to 96-well plates and mixed with 100 μL modified Griess reagent (Invitrogen, Carlsbad, CA, USA). The absorbance of each well was measured at 540 nm using a Versa max plate reader (Molecular Devices). A standard curve with increasing sodium nitrite concentrations was generated in parallel and used for quantification.

### 4.10. Measurement of GI Hormone Concentrations

The concentrations of CCK and gastrin were quantified using ELISA kits (Cusabio Biotech Co., Ltd., Wuhan, China), according to the manufacturer’s instructions. Briefly, mid colon tissues (50 mg) from mice (*n* = 5) in subset group were homogenized in ice-cold 1× PBS (pH 7.2–7.4) using a glass homogenizer (Sigma-Aldrich Co.). The resultant tissue lysates were then centrifuged at 1000× *g* for 5 min at 4 °C, after which the supernatant was collected for analysis. After adding the two specific hormone antibodies (separately in each well), the supernatant was incubated for 1 h at 37 °C, to which HRP–streptavidin solution was then added and incubated further for 1 h at 37 °C. This was followed by adding the TMP One-Step Substrate Reagent, followed by incubation of the mixture for 30 min at 37 °C. The reaction was terminated by adding the stop solution. Finally, the absorbance of the reaction mixture was read at 450 nm using the Molecular Devices VersaMax Plate Reader (Sunnyvale, CA, USA).

### 4.11. Statistical Analysis

The statistical significance was evaluated using one-way analysis of variance (ANOVA) (SPSS for Windows, Release 10.10, Standard Version, Chicago, IL, USA), followed by Tukey post hoc *t*-test for multiple comparisons. All values are expressed as the means ± SD, and *p*-values < 0.05 were considered significant.

## 5. Conclusions

This study investigated whether C3 deficiency-induced constipation is accompanied by alterations in the excitatory and inhibitory function of ENS in the mid colon of C3 KO mice. The present study first suggests that C3 deficiency-induced constipation is associated with a decrease in the density of enteric neurons and ICC, a decline in 5-HT and ACh concentration, and an increase in the NO concentration for ENS function in the mid colon. In addition, the results suggest that this type of constipation is associated with the upregulation of C3aR and C3bR expression during the complete downregulation of the C3 protein. Furthermore, these results provide novel evidence of the potential for therapeutic targets to prevent and treat patients with C3-related constipation.

## Figures and Tables

**Figure 1 ijms-23-06862-f001:**
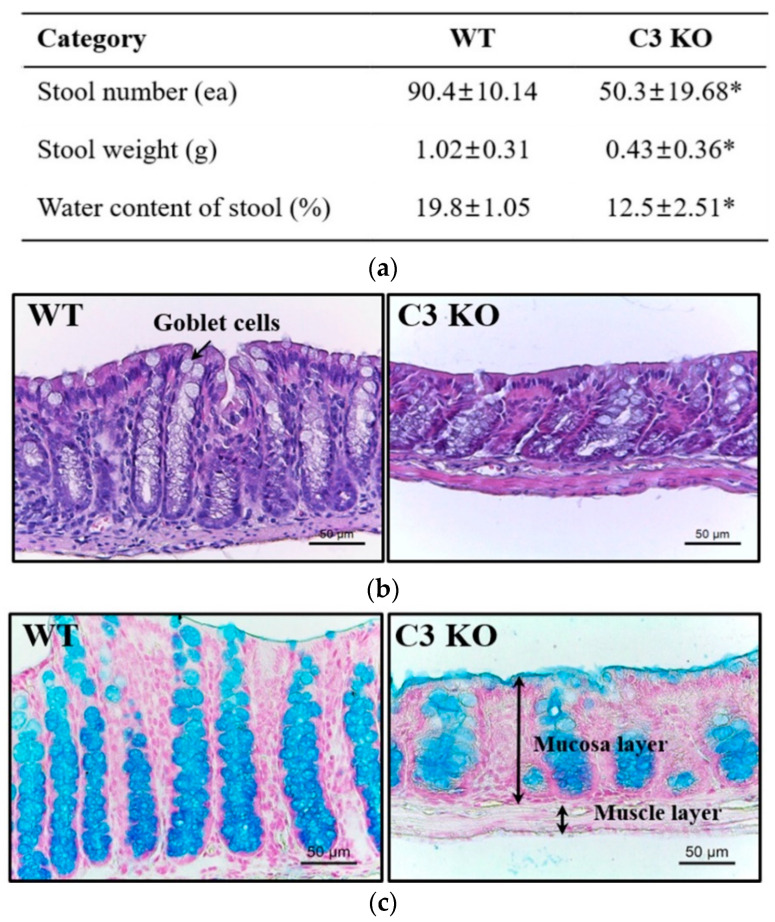
Determination of the constipation phenotypes in C3 KO mice. (**a**) Stool parameters. After collecting the stools from a metabolic cage, their number, weight, and water contents were measured, as described in Section 4. Seven mice per group were used for stool collection, and each parameter was assayed in duplicate. The data are reported as the mean ± SD values. * *p* < 0.05 compared with the WT group. (**b**) Histological structure of the mid colon. H&E-stained sections of the mid colon from WT and KO mice were observed at 100× and 400× magnification using a light microscope. Five mice per group were used to prepare the H&E-stained slides, and the histopathological structures were observed in duplicate in each slide. (**c**) Mucin staining of the mid colon. Alcian blue-stained sections of the mid colon from WT and KO mice were observed at 100× and 400× magnification using an optical microscope. Three mice per group were used to prepare Alcian blue-stained slides, and density changes were observed in duplicate in each slide. Abbreviations: WT, wildtype; KO, knockout; H&E, hematoxylin and eosin.

**Figure 2 ijms-23-06862-f002:**
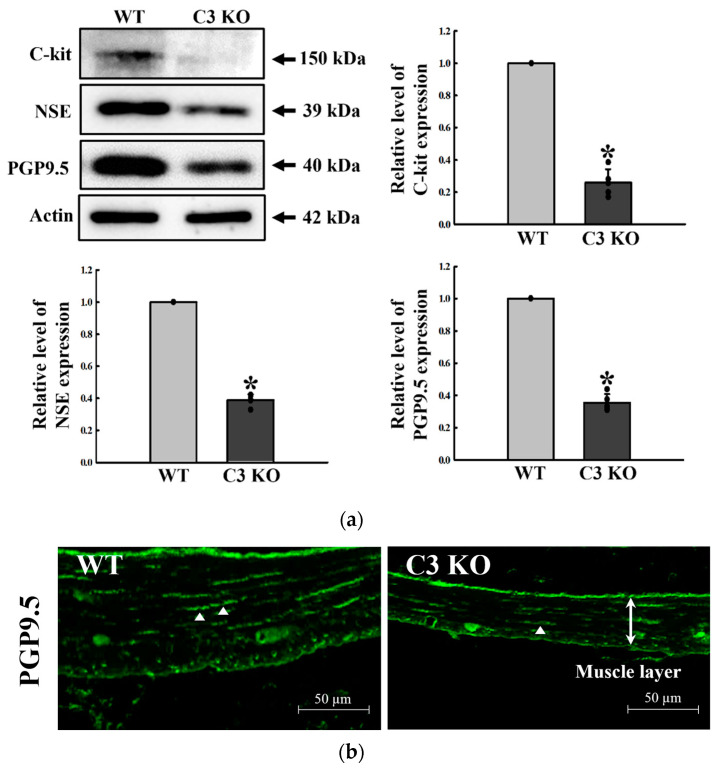
Expression levels of C-kit, NSE, and PGP9.5 in the mid colon of C3 KO mice. (**a**) Expression levels of C-kit, NSE, and PGP9.5 proteins. The expression level of the three proteins was determined by Western blot analysis using the specific primary antibody and HRP-labeled anti-rabbit IgG antibody. The band intensities were determined using an imaging densitometer, and protein expressions were calculated relative to the intensity of β-actin. Four mice per group were used to prepare the tissue lysates, and Western blots were assayed in duplicate for each sample. Data are reported as the mean ± SD. * *p* < 0.05 compared with the WT group. (**b**) Tissue distribution of PGP9.5 proteins. This level was detected in the mid colon using an immunofluorescence (IF) staining assay. Three mice per group were used in the slide section, and IF staining was assessed in duplicate in two different slides. Arrowheads indicate PGP9.5-stained subpopulations. Abbreviations: WT, wildtype; KO, knockout; C-kit, receptor tyrosine kinase; NSE, neuron-specific enolase; PGP9.5, protein gene product 9.5; HRP, horseradish peroxidase; IgG, immunoglobulin G.

**Figure 3 ijms-23-06862-f003:**
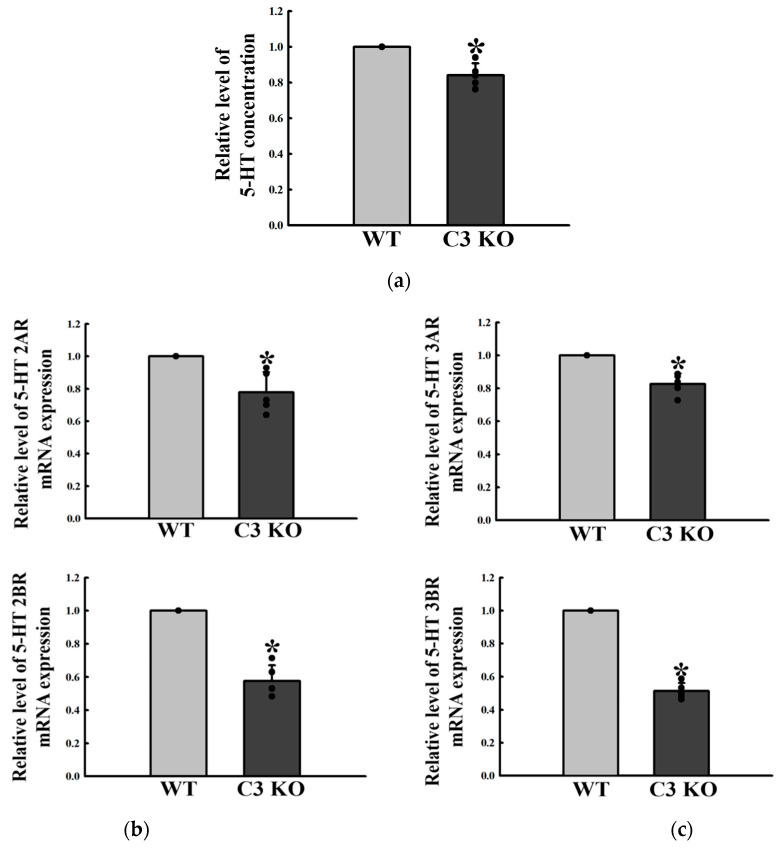
Levels of 5-HT and its receptors in the mid colon of C3 KO mice. (**a**) 5-HT concentration. The concentration of 5-HT was measured in the mid colon homogenate using an enzyme-linked immunosorbent assay. The minimum detectable concentration of each kit was 1.5–250 ng/mL of 5-HT. Five mice per group were used to prepare tissue homogenate, and the 5-HT concentration was assayed in duplicate. (**b**) Expression of 5-HT 2AR and 2BR. The levels of 5-HT 2AR and 2BR mRNA expression in the total mRNA of colon tissue were measured by qRT-PCR analyses using the specific primers. The mRNA level of each gene was calculated on the basis of the intensity of β-actin as an endogenous control. (**c**) Expression of 5-HT 3AR and 3BR. The levels of 5-HT 3AR and 3BR mRNA expression in the total mRNA of colon tissue were measured by qRT-PCR analyses using the specific primers. The mRNA level of each gene was calculated on the basis of the intensity of β-actin as an endogenous control. Four mice per group were used to prepare the tissue homogenate, and the total RNA, 5-HT concentration, and qRT-PCR analysis were assayed in duplicate for each sample. Data are reported as the mean ± SD. * *p* < 0.05 compared to the WT group. Abbreviations: WT, wildtype; KO, knockout; 5-HT, 5-hydroxytryptamine; qRT-PCR, quantitative real-time polymerase chain reaction; 5-HT 2AR, 5-hydroxytryptamine 2A receptor; 5-HT 2BR, 5-hydroxytryptamine 2B receptor; 5-HT 3AR, 5-hydroxytryptamine 3A receptor; 5-HT 3BR, 5-hydroxytryptamine 3B receptor.

**Figure 4 ijms-23-06862-f004:**
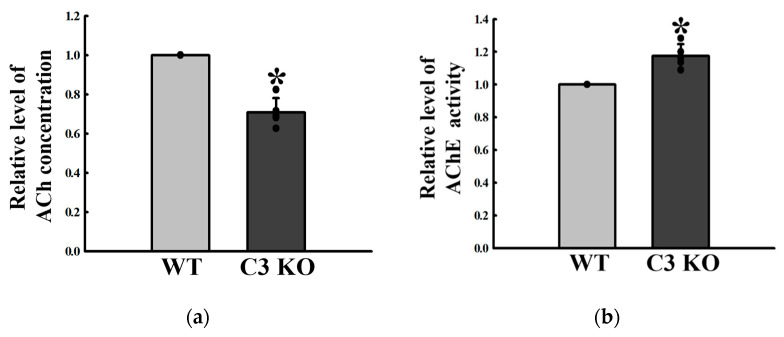
ACh concentrations and AChE activity in the mid colon of C3 KO mice. (**a**) ACh concentrations. The concentration of ACh was measured in the mid colon homogenate using an Acetylcholine Assay Kit. The minimum detectable concentration of this kit is 0.078–20 µM of ACh. (**b**) AChE activity. After homogenizing the mid colon tissue, the AChE activity was measured using an Acetylcholinesterase Assay Kit that could detect as little as 0.01 mU AChE in a 100 µL assay volume (0.1 mU/mL). Five mice per group were used to prepare tissue homogenate, and the ACh concentration and AChE activity were assayed in duplicate. Data are reported as the mean ± SD. * *p* < 0.05 compared to the WT group. Abbreviations: WT, wildtype; KO, knockout; ACh, acetylcholine; AChE, acetylcholinesterase.

**Figure 5 ijms-23-06862-f005:**
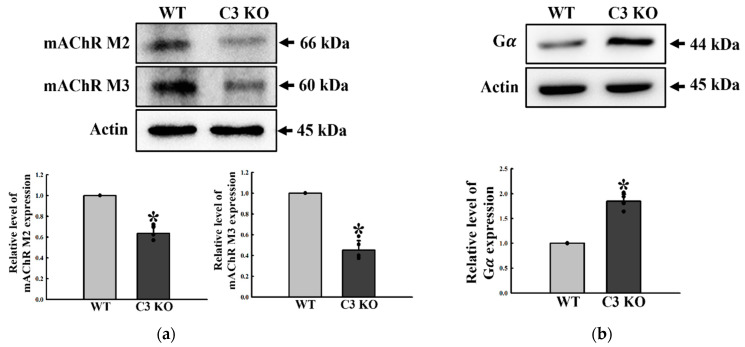
Expressions of mAChRs and key mediators within mAChRs downstream signaling pathway in the mid colon of C3 KO mice. (**a**) mAChR M2 and M3 expression. The expression levels of mAChR M2 and M3 were measured by Western blot analysis using the specific primary antibodies and HRP-labeled anti-rabbit IgG antibody. (**b**) Gα expression. The expression levels of Gα in the mAChR M2 and M3 downstream signaling pathway were measured by Western blot analysis using the specific primary antibodies and HRP-labeled anti-rabbit IgG antibody. (**c**) PKC, p-PKC, PI3K, p-PI3K, MLC, and p-MLC expression. The expression levels of six proteins in the mAChR M2 and M3 downstream signaling pathway were measured by Western blot analysis using the specific primary antibodies and HRP-labeled anti-rabbit IgG antibody. After the intensity of each band was determined using an imaging densitometer, the relative levels of the four proteins were calculated on the basis of the intensity of β-actin. Four mice per group were used to prepare the total tissue homogenate, and Western blot analyses were assayed in duplicate. Data are reported as the mean ± SD. * *p* < 0.05 compared to the WT group. Abbreviations: WT, wildtype; KO, knockout; mAChR, muscarinic acetylcholine receptors; Gα, G-protein alpha subunit; PKC, protein kinase C; PI3K, phosphoinositide 3-kinase; MLC, myosin light chain; HRP, horseradish peroxidase; IgG, immunoglobulin G.

**Figure 6 ijms-23-06862-f006:**
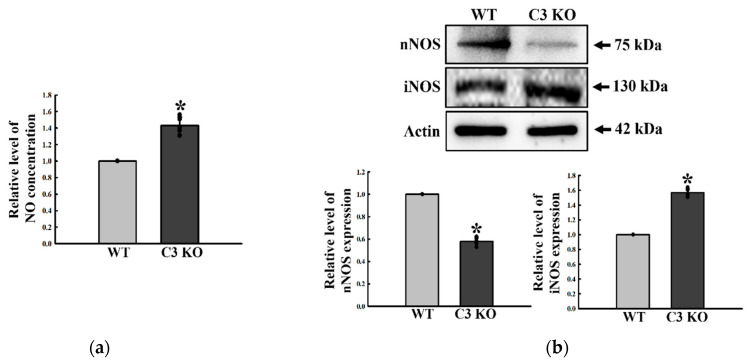
Levels of NO, nNOS, and iNOS in the mid colon of C3 KO mice. (**a**) NO concentration. The concentration of NO was determined using the Griess reagent. Five mice per group were used to prepare the tissue lysates, and the NO concentration was assayed in duplicate for each sample. (**b**) nNOS and iNOS expression. The expression levels of the nNOS and iNOS proteins were determined by Western blot analysis using the specific primary antibody and HRP-labeled anti-rabbit IgG antibody. The band intensities were determined using an imaging densitometer, and protein expressions were calculated relative to the intensity of β-actin. Four mice per group were used to prepare the tissue lysates, and Western blots were assayed in duplicate. Data are reported as the mean ± SD. * *p* < 0.05 compared to the WT mice. Abbreviations: WT, wildtype; KO, knockout; NO, nitric oxide; nNOS, neuronal nitric oxide synthase; iNOS, inducible nitric oxide synthase; HRP, horseradish peroxidase; IgG, immunoglobulin G.

**Figure 7 ijms-23-06862-f007:**
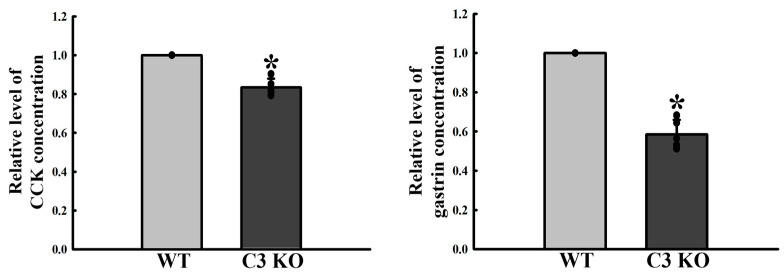
Concentrations of GI hormones. The concentrations of CCK and gastrin were measured in the mid colon homogenate using an enzyme-linked immunosorbent assay. The minimum detectable concentration of each kit was 0.1–1000 pg/mL of CCK and 0.312–20 pg/mL of gastrin. Five mice per group were used to prepare tissue homogenate, and the hormone levels were assayed in duplicate for each sample. Data are reported as the mean ± SD. * *p* < 0.05 compared to the WT group. Abbreviations: WT, wildtype; KO, knockout; GI hormone, gastrointestinal hormone; CCK, cholecystokinin.

**Figure 8 ijms-23-06862-f008:**
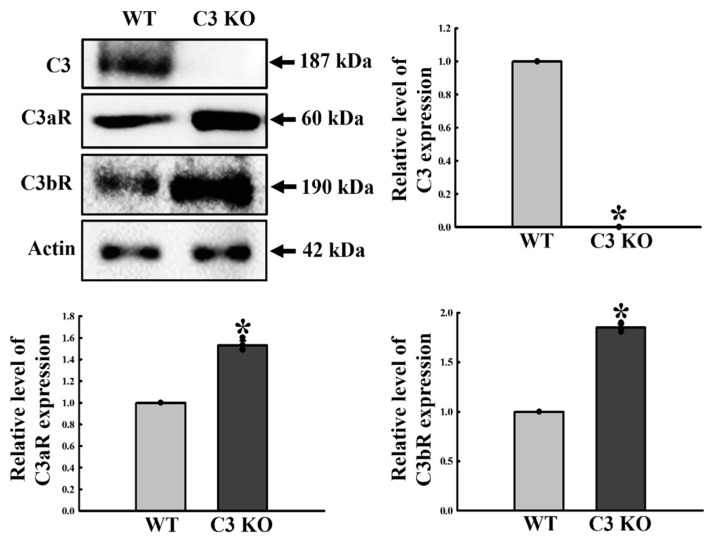
Expression of C3aR and C3bR proteins in the mid colon of C3 KO mice. The expression levels of the C3, C3aR, and C3bR proteins in the mid colon were measured by Western blot analysis using the anti-C3aR and anti-C3bR antibodies. After determining the intensity of each band using an imaging densitometer, the relative levels of the C3, C3aR, and C3bR protein were calculated on the basis of the intensity of β-actin. Four mice per group were used to prepare the tissue lysates, and Western blots were assayed in duplicate for each sample. Data are reported as the mean ± SD. * *p* < 0.05 compared to the WT group. Abbreviations: WT, wildtype; KO, knockout; C3aR, C3a receptor; C3bR, C3b receptor; HRP, horseradish peroxidase; IgG, immunoglobulin G.

## Data Availability

All the data that support the findings of this study are available on request from the corresponding author.

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
