# Peer review of "Dysregulation of the Enteric Nervous System in the Mid Colon of Complement Component 3 Knockout Mice with Constipation Phenotypes"

_ijms, 2022, doi:10.3390/ijms23126862_

Round 1

Reviewer 1 Report

The study entitled "Dysregulation of the enteric nervous system in the mid colon of 2 Complement component 3 knockout mice with constipation 3 phenotypes" describes alterations in the expression of several neuronal, hormonal and myogenic markers that occur in the complement component 3 knockout mouse model of constipation. The results are generally convincing but the manuscript can be improved by reformatting of graphs to show individual values, and through minor revisions to phrasing and terminology.

Comments:

Figure 2(a), Figure 3(a-c), Figure 4 (a-b), Figure 5 (a-c), Figure 6 (a-b), Figure 7, Figure 8 : Show individual data points; not just bar graphs. Using the types shown in Figure 1B-C here: https://www.ahajournals.org/doi/10.1161/CIRCULATIONAHA.118.037777 will significantly improve the manuscript.

Figure 1(a): For ease of comparison between studies, also report the hourly stool output

Figure 1(a): Accounting for all 3 parameters, the data suggests that water volume lost per hour in stool from C3 KO mice (0.225mL/hour) was ~15% of the amount of water excreted via stool from WT mice (1.52mL/hour). In a previous study you have shown no significant differences in food/water intake in C3 KO mice, compared to WT (Park et al., 2021). Does this imply C3 KO mice show increased urination volumes? How is the extra water excreted in C3 KO mice?

Figure 2(b): Indicate the significance of the arrowheads in the figure legend

Line 290: reference the conflicting results

Discussion, there are several instances of using the word “remarkably” rather than “statistically significantly” results. Correct this

Analysis of markers associated with ENS excitatory or inhibitory transmission

Lines 128, 131, 190, 192, 255, 276, 510: should be something like “alterations in markers associated with excitatory/inhibitory transmission in the ENS”. In their current form, the terminology may give readers the impression functional studies were conducted. Lines 22 and 77, 80 are phrased more accurately.

Line 261: or across the whole colon

Line 252: “was” rather than “can be”

Line 194: “increased by 41%” rather than “remarkably higher at 41%”

Line 195: “magnitude”, “degree” or “amount” rather than “the rate”

Line 21: “the changes in expression of neuronal and ICC markers”. In general make clear the study assessed changes in the expression of neuronal and ICC markers, NOT the number neurons and ICC. It is fine to speculate that loss of neurons and ICC underlie the changes in expression of their markers.

Line 20-21: “To determine the role of C3 in regulating the ENS during C3 deficiency-induced constipation…” rather than “To determine the role of C3 in the regulatory mechanism of ENS during C3 deficiency-induced constipation”

Line 36: “dysregulation of the ENS”, rather than “dysregulation of ENS”

Ref:

Park JW, Kim JE, Choi YJ, Kang MJ, Choi HJ, Bae SJ, Hong JT, Lee H, Hwang DY (2021) Deficiency of complement component 3 may be linked to the development of constipation in FVB/N-C3(em1Hlee) /Korl mice. The FASEB Journal 35:e21221.

Reviewer 2 Report

This study is the continuation of the previous and recently published studies by the Authors:

– Park, J.W.; Kim, J.E.; Choi, Y.J.; Kang, M.J.; Choi, H.J.; Bae, S.J.; Hong, J.T.; Lee, H.; Hwang, D.Y. Deficiency of complement component 3 may be linked to the development of constipation in FVB/N-C3em1Hlee/Korl mice. FASEB J. 2021, 35, e21221.

– Choi, Y.J.; Kim, J.E.; Lee, S.J.; Gong, J.E.; Son, H.J.; Hong, J.T.; Hwang, D.Y. Dysbiosis of fecal microbiota from complement 3 knockout mice with constipation phenotypes contributes to development of defecation delay. Front. Physiol. 2021, 12, 650789.

– Choi, Y.J.; Kim, J.E.; Lee, S.J.; Gong, J.E.; Jin, Y.J.; Lee, H.; Hwang, D.Y. Promotion of the inflammatory response in mid colon of complement component 3 knockout mice. Sci. Rep. 2022, 12, 1-17.

aiming to characterise the complement component C3 knockout mice. In terms of the constipation phenotype, the characterisation of the changes to the enteric nervous system and the interstitial cells of Cajal  in the colon is an obvious choice, which justifies the aim of the submitted article. However, based on the methodology, which was used in the study, false and premature conclusions are made. While, in fact, the study is incomplete. In the discussion section, the Authors wrote: "further studies will be needed to directly count and identify the number of enteric neurons in the ICC after staining with the specific antibody”. Unfortunately, one cannot draw any conclusions related to the enteric neurons and ICCs without their full immunohistochemical characterisation and based only on quantitative methods such as Western blotting (at the same time in the cited sentence there is a mistake: ”enteric neurons AND ICCs” instead of „enteric neurons IN THE ICC” - such mistakes are in fact abundant throughout the whole manuscript). And this is the reason why I think it is premature to publish the manuscript.

Besides, there are numerous other issues, which should be deeply corrected. Below, I’ll pinpoint SOME (not all) most obvious ones:

  • „Sixteen-week-old WT (n=7) and KO (n=7) mice were bred in a metabolic cage 360 to analyze the excretion parameters” - yet in most of the analyses, in the results section the following information is given: „Three to five mice per group were used to prepare…” - so either 3 or 5 - exact numbers should be given, as „3” is relatively low and drawing conclusions based on statistics on 3 animals may lead to false conclusions, and should be added to the limitations of the study,
  • vendors’ numbers of the used antibodies should be given so one could compare the results in the future studies (vendors’ names are not enough!),
  • why the mid colon was decided to be used - no argument is given,
  • in case of the neuronal marker PGP 9.5 - the IF was performed however there was no proper morphometrical analysis done, but only the measurement of the level of fluorescence (according to the information given in the methods section), which was further given only as „lower” - „The level of PGP9.5 expression 111 was confirmed by morphometric analysis of the neural population. The PGP9.5-stained 112 subpopulation was lower in the mid colon of C3 KO mice than in WT mice”. Besides, as the ganglia of the ENS are packed between the submucosal layer and muscle layer and between the muscle layers, obviously one will expect less amount of the PGP 9.5 protein in the knockout mice since their muscle layer was much more narrow in comparison with the wild type. And this is why, for example,  WB results should be accompanied by the detailed (also in therms of the description of the methodology - how many sections and how many fields of vision were analysed, etc…) morphometric analysis of the staining, 
  • c-kit is a marker of other than ICCs cells as well, thus WB quantitative analysis of the colon homogenates thus not equal the amount of the ICCs in the colonic wall!
  • in the discussion section, limitations of the study should be clearly discussed,
  • the discussion section is very premature, as any comparisons should be made in relation to other animal models with similar phenotypes only, not clinical observations due to the fact that human disease ethiopathogenesis is much more complex, and especially based on the given (limited!) results.

Reviewer 3 Report

This research article is focused on understanding the alteration of the regulatory markers for the ENS function in the mid colon of C3 KO mice with the constipation phenotypes.  The paper is well written and easy to follow. The existing literature for undergoing this study is well cited. I have some comments that the authors should address:

LINE 135- change 3BR to 2BR
LINE 156 and LINE 170 and for all the other experiments- Why is there no specific n for each analysis- on what basis was it decided and was it powered enough.
LINE 155- there is growing evidence suggesting at least 3 HK genes to use as controls. please discuss why only one HK gene
LINE 337- authors should discuss why the results were opposite to the available literature and how future studies can help answer this.

I would have assumed the authors would discuss the limitations of the study in particular about not using sex-specific models or the data normalization using only one HK gene.

Authors should include how the q-PCR data was normalized- geomean or arithmetic mean was used for the livak method.

Round 2

Reviewer 2 Report

Unfortunately, not all the points raised in my previous review were addressed.

Reviewer 3 Report

No comments

Author Response

Thank you.

Round 3

Reviewer 2 Report

The manuscript can be accepted in the present form.